# Ripples in the bottom of the potential energy landscape of metallic glass

Leo Zella[1], Jaeyun Moon [2,4] & Takeshi Egami [1,2,3] ✉

In the absence of periodicity, the structure of glass is ill-defined, and a large number of structural states are found at similar energy levels. However, little is known about how these states are connected to each other in the potential energy landscape. We simulate mechanical relaxation by molecular dynamics for a prototypical $Cu_{64.5}Zr_{35.5}$ metallic glass and follow the mechanical energy loss of each atom to track the change in the state. We find that the energy barriers separating these states are remarkably low, only of the order of 1 meV, implying that even quantum fluctuations can overcome these potential energy barriers. Our observation of numerous small ripples in the bottom of the potential energy landscape puts many assumptions regarding the thermodynamic states of metallic glasses into question and suggests that metallic glasses are not totally frozen at the local atomic level.

It is immensely difficult to characterize the relationship between structure and dynamics in disordered matters such as liquids and glasses. Conceptually, at least, the complex states and dynamics can be described through the potential energy landscape (PEL)[1–4]. When a liquid is cooled without a phase change into a crystal, it falls into a local minimum in the PEL and becomes a glass, trapped in a non-equilibrium state. Given sufficient thermal and mechanical energies, transitions from this local minimum to other local minima occur through activation and relaxation processes over energy barriers. Classical depictions of the PEL focus on basins and metabasins which includes multiple basins, with the $\alpha$-relaxation to connect different metabasins and the $\beta$-relaxation to connect basins. Within the basin glasses are assumed to be 'frozen'. However, recent simulation[5,6] and theory[7] suggest a more complex picture. The basins and metabasins in the PEL form a hierarchical structure, allowing for a greater diversity of activation-relaxation phenomena. Since these features in the PEL play a crucial role in the structure-property relationship in disordered matters, it is important to characterize features of the basins in the PEL and how they are inter-connected.

These features within the basins should manifest as relaxation processes which can be probed mechanically. The $\alpha$-relaxation and $\beta$-relaxation have been extensively studied using standard mechanical techniques such as dynamic mechanical spectroscopy (DMS)[8,9] and atomic force microscopy[10–12]. An early simulation work[13] suggested that

metallic glasses undergo atomic-level plastic deformation even below the yield point. Because every atom in metallic glasses has a different local environment, when a macroscopic strain is applied across the system, there arise non-uniform strains at the atomic-level[14]. Such non-uniform strains could result in locally plastic behavior even in the apparently elastic regime. This prediction is now supported by a growing body of experimental works showing atomic-level local deformation upon loading and low energy mechanical relaxation in the form of microscopic rearrangements below the yield point[14–16], and also by simulation works demonstrating microscopic rearrangements in the apparently elastic regime[13,17–19].

To probe the microscopic plastic behavior, we utilize molecular dynamics dynamic mechanical spectroscopy (MD-DMS) on a prototypical $Cu_{64.5}Zr_{35.5}$ metallic glass ($T_g \sim 700$ K). We apply uniform sinusoidal shear strains with the maximum amplitude of $\epsilon_A = 1.0\%$ and 2.5 % and a period of 100 ps at temperatures from 0.1 K to 300 K and decompose the stress into atomic-level stresses to observe the atomic response to a mechanical perturbation. We perform a cycle-to-cycle analysis to determine the atomic-level phase shift of the atomic-level shear stress compared to the macroscopic strain and evaluate the atomic-level energy loss due to atomic rearrangements. We find that above 100 K, most lossy atoms identified in one shear deformation cycle convert to non-lossy behavior in the next cycle, demonstrating high transience of local atomic structure participating in mechanical

[1]Department of Materials Science and Engineering, The University of Tennessee, Knoxville, TN 37996, USA. [2]Materials Science and Technology Division, Oak Ridge National Laboratory, Oak Ridge, TN 37831, USA. [3]Department of Physics and Astronomy, The University of Tennessee, Knoxville, TN 37996, USA. [4]Present address: Sibley School of Mechanical and Aerospace Engineering, Cornell University, Ithaca, NY 14853, USA. ✉e-mail: egami@utk.edu

relaxation. In contrast, at cryogenic temperatures lossy atoms are largely reversible. From this temperature dependence, we find the activation energy for local structural change to be quite small, only on the order of $E_A \approx 1$ meV. Given that these activation energies are well below those of the $\alpha$ and $\beta$ relaxation, we ascribe this to the small ripples in the bottom of the basin in the PEL. Such ripples were largely overlooked in earlier studies, but they may be related to the nearly constant loss observed at low temperatures[20,21]. Additionally, we estimate the magnitude of quantum fluctuations using the vibrational density of states. The zero-point vibration energy is far above the energy barriers of the ripple, implying that quantum fluctuations easily connect multiple subbasins. Our results suggest that the system moves easily within the PEL basin through quantum fluctuations and thermal excitations and challenges many assumptions regarding the thermodynamic states of metallic glasses.

## Results

### Lossy atom distribution

In an earlier study, we have demonstrated that the individual atomic-level shear stress is a powerful tool to identify the subset of atoms in the system responsible for the mechanical loss in a single cycle of shear strain (see Methods for details)[22]. From the atomic-level shear stress response in frequency space (Fig. 1a, b), we estimate individual atomic-level phase shift $\delta_{atom}$, which corresponds to the atomic-level mechanical loss. The calculated $\delta_{atom}$ for all atoms over a single cycle of shear strain shows a nearly symmetric distribution of atomic-level phase shift (Fig. 1c). We define atoms directly involved in the mechanical loss during the strain cycle by an atomic-level phase of $\delta_{atom} \in [\frac{\pi}{4}, \frac{3\pi}{4}]$. These lossy atoms are shown in the cross-hatched region in Fig. 1c. The identification of the phase of lossy atoms during mechanical loss was shown in a previous work[22].

By examining if atoms identified as lossy at a cycle (the cycle $n$) are still lossy in the subsequent cycle (the cycle $n+1$) we can study how reversible the mechanical loss is at the atomic level. We find that the result strongly depends on temperature, but not much on cooling rate. We see that at a cryogenic temperature of 0.1 K, the distribution of $\delta_{atom}(n+1)$ of lossy atoms remains centered around $\delta_{atom} \approx \frac{\pi}{2}$, as shown in Fig. 2, indicating most lossy atoms remain lossy. However, the distribution of $\delta_{atom}(n+1)$ changes dramatically with a relatively small increase in temperature, with the peak at $\delta_{atom} \approx \frac{\pi}{2}$ decreasing rapidly in height and creating a distribution more akin to those of the whole system, centered around $\delta_{atom} \approx 0$, as seen in Fig. 1c. This rapid change represents a change from mostly reversible mechanical loss at very low temperatures to irreversible/transient mechanical loss at elevated temperatures. At higher temperatures, atoms identified as lossy in the first cycle are more likely to be no longer lossy and part of the elastic peak at $\delta_{atom} \approx 0$ in the next cycle. Comparing these temperature dependent trends for samples prepared with different cooling rates, represented in Fig. 2a–d, minimal differences are seen over the four decades of cooling rates.

### Temperature dependence of lossy fraction

To understand the evolution of lossy atoms over many cycles, we calculate the fraction of atoms identified as lossy at the cycle $n$, and remain lossy after additional $m$ cycles, as $\langle f_{lossy}(n, n+m)\rangle$ (see Eq. (2) in Methods section). Over the 30 cycles in which lossy atoms were identified, the fraction of lossy atoms that remain lossy for up to 9 cycles later and across a variety of temperatures and cooling rates is shown in Fig. 2. By definition, the $\langle f_{lossy}\rangle$ fraction starts at 1 for $m=0$, that is the cycle when lossy atoms were identified. As shown in Fig. 3, there is a large drop in the fraction with increasing $m$. Interestingly, the fraction decays significantly at the first additional cycle ($m=1$) and does not decay much afterwards ($m=2-9$), apart from the sample with the highest cooling rate which shows a slower decay, at intermediate temperatures (10–75 K) seen in Fig. 3a. For all cooling rates, $\langle f_{lossy}\rangle$ at 0.1 K remains high at around 0.7 even up to 9 cycles later,

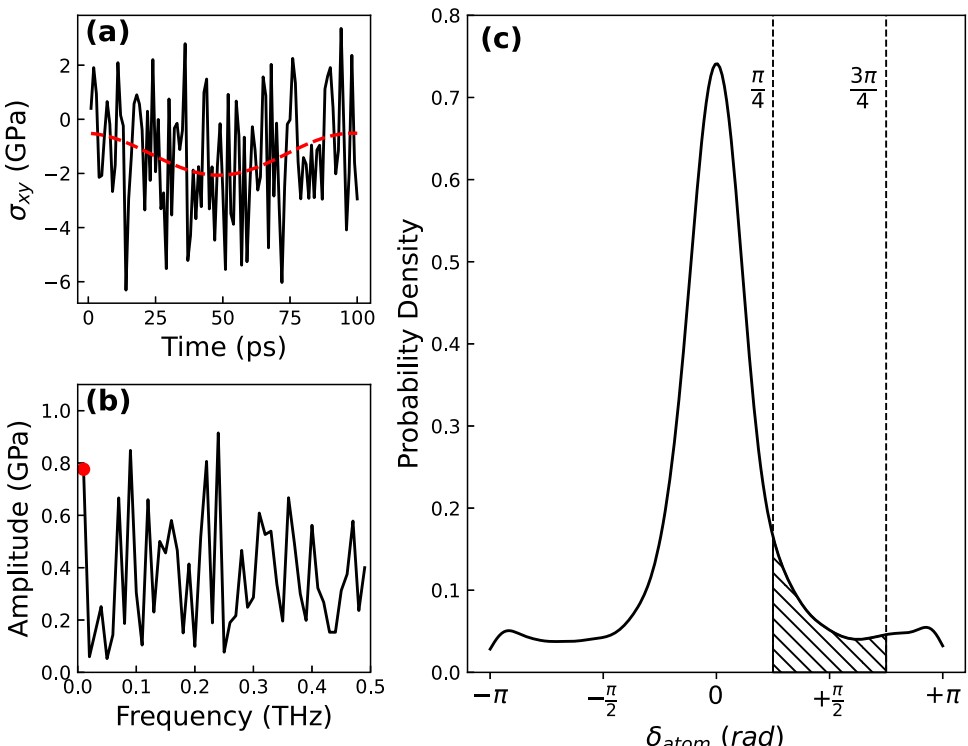

**Fig. 1 | Method for calculating atomic-level phase shift in frequency space and the resulting distribution of atomic-level phase shift at 300 K for the slowest cooling rate of 0.01 K/ps. (a)** Atomic-level shear stress of a lossy atom with red dashed line representing amplitude and phase shift of atom calculated by Fourier transform. **(b)** Amplitude response in frequency representation of atomic-level shear stress with red data point representing the sinusoidal shearing frequency. **(c)** Atomic-level phase shift ($\delta_{atom}$) distribution for all atoms from 1 cycle at 300 K. The hatched area within dashed lines indicates the bounds of the definition for lossy atoms.

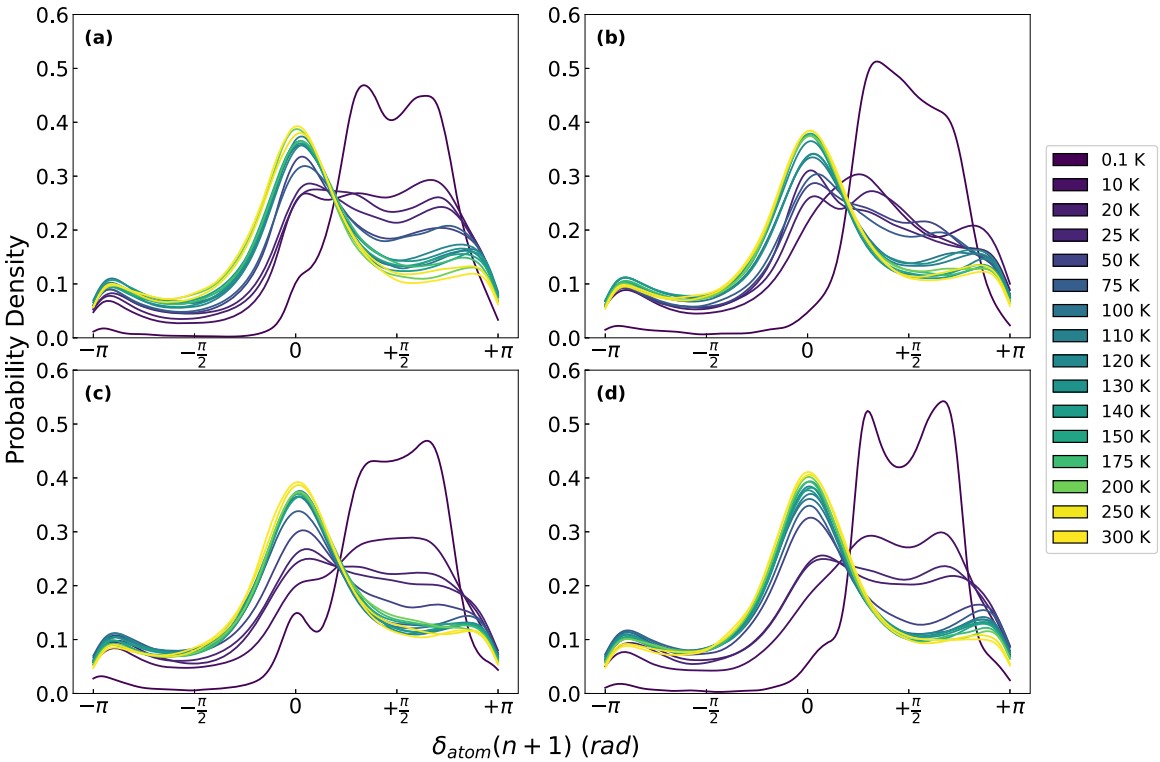

**Fig. 2 | Lossy atom phase shift distribution 1 cycle after initial identification of lossy atoms at the strain of 2.5%.** It is shown that at low temperatures, the mechanical relaxation is mostly reversible with lossy atoms remaining lossy (that is with $\delta_{atom} \approx \pi/2$). As temperature increases, the distribution more closely resembles that of the distribution of all atoms for all cooling rates at (**a**) 100 K/ps, (**b**) 1 K/ps, (**c**) 0.1 K/ps, and (**d**) 0.01 K/ps.

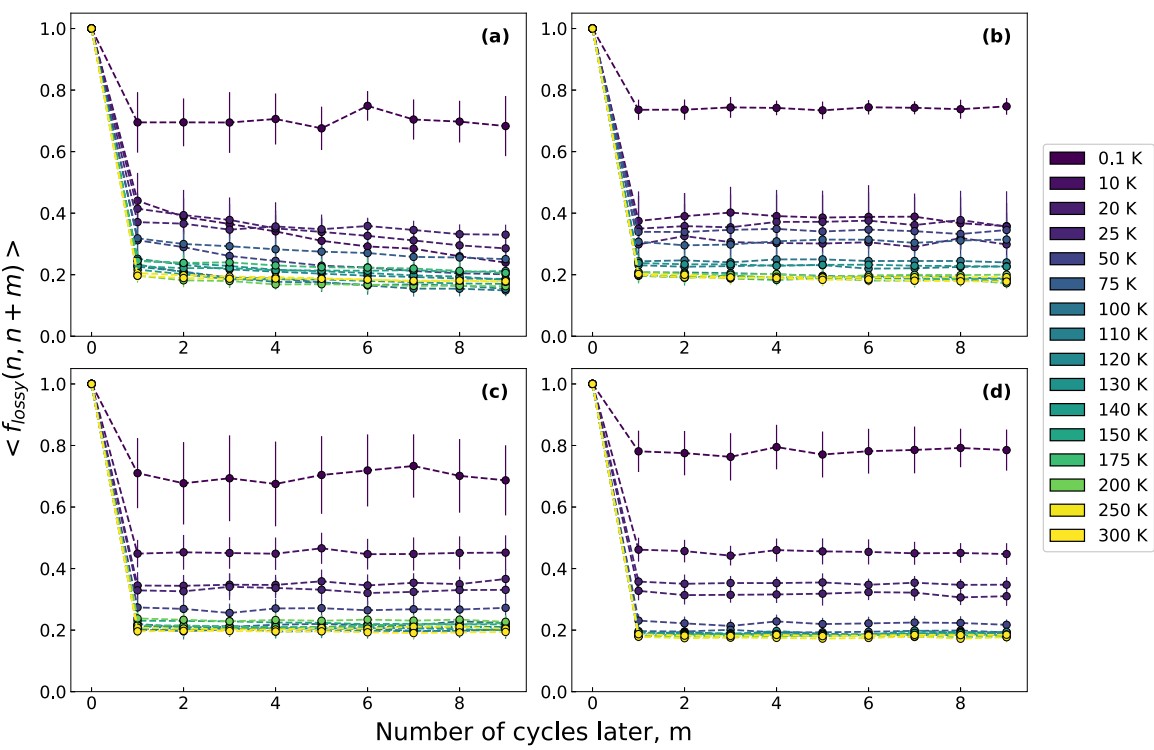

**Fig. 3 | The average fraction of lossy atoms that remain lossy using a strain of 2.5 %, $\langle f_{lossy}(n, n+m) \rangle$, where $m$ is the number of cycles later from cycle of identification $n$.** Data is averaged from 20 cycles with error bars representing the standard deviation. This is done for a variety of cooling rates at (**a**) 100 K/ps, (**b**) 1 K/ps, (**c**) 0.1 K/ps, and (**d**) 0.01 K/ps.

indicating lossy atoms remain largely lossy in a largely reversible, non-transient manner at $T = 0.1$ K. Remarkably, the $\langle f_{lossy} \rangle$ fraction is rapidly reduced with the increase in temperature until reaching a plateau of $\langle f_{lossy} \rangle \approx 0.2$ at around 100 K. This represents a characteristic change in mechanical loss from highly reversible, non-transient behavior at cryogenic temperatures to highly transient behavior at temperatures above 100 K.

In order to study the role of the magnitude of strain, we plot the $\langle f_{lossy}(n, n+1) \rangle$ for just one cycle after identification at a shear strain of $\epsilon_A = 2.5\%$ and $\epsilon_A = 1\%$, both of which are in the elastic regime macroscopically as shown in Supplementary Fig. 1. The plot of $\langle f_{lossy}(n, n+1) \rangle$ in Fig. 4 clearly shows the same massive decrease in the lossy fraction as a function of temperature for both strain magnitudes. Interestingly, for the strain of 1 %, we observe that the transition in temperature dependence to the plateau occurs at a lower temperature than for 2.5 % strain. To further confirm that the large drop in lossy fraction occurs mostly after the first cycle, we also calculated $\langle f_{lossy}(n, n+9) \rangle$, that is the fraction of lossy atoms that remained lossy 9 cycles later, as seen in Supplementary Fig. 2, which largely demonstrates the same trend.

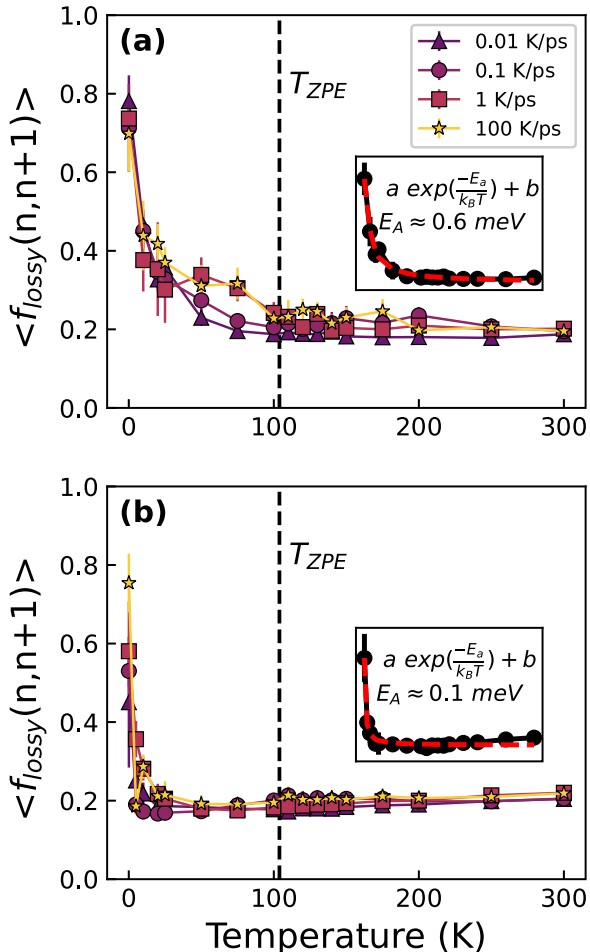

**Fig. 4 | The average fraction of lossy atoms 1 cycle after initial identification at a strain of 2.5% and 1.0%.** Lossy fraction $\langle f_{lossy}(n, n+m) \rangle$ for various cooling rates demonstrating the rapid transition from mostly reversible to mostly irreversible mechanical loss using (**a**) $m = 1$ (one cycle later) at the strain of 2.5% with inset figure showing exponential fit and (**b**) $m = 1$ (one cycle later) at the strain of 1%. For both strains, a rapid transition from low transience where lossy atoms mostly remain lossy to high transience is shown with emerging plateau at high temperatures. The dashed line represents the effective zero-point vibration temperature which is in the region of high transience. Inset figure shows exponential fit to the slowest cooled system of 0.01 K/ps and gives the calculated activation energy. Data is averaged from 20 cycles with error bars representing the standard deviation.

To characterize the temperature dependence of the lossy fraction, we fit an exponential function as $\langle f_{lossy}(n,n+1) \rangle \approx a e^{-\frac{E_A}{k_B T}} + b$ to calculate the apparent activation energy of the atomic rearrangements related to the change in mechanical loss. We find an apparent activation energy of $E_A \approx 0.6$ meV at $\epsilon_A = 2.5\%$ and $E_A \approx 0.1$ meV at a strain of $\epsilon_A = 1\%$, with fit seen in the inset of Figs. 4a, b for the slowest cooling rate. These activation energies are significantly smaller than those for the typical $\beta$ relaxation in metallic glasses which is on the order of 1 eV[23,24]. From the use of two different strain magnitudes, we observe a similar rapid change from reversible to mostly irreversible and transient mechanical relaxation for two different strain magnitudes.

As this rapid change in mechanical loss happens at low temperatures with such low activation energies, we next examined if quantum mechanical zero-point fluctuation is sufficient to overcome such small barriers. Utilizing the vibrational density of states (vDOS) of the system calculated at 0.1 K we calculate the total vibrational ground state energy to be 26.8meV/atom. By equating this ground state energy to the total thermal energy of the system we estimate an effective temperature of the zero-point vibrational motion, $T_{ZPE}$, to be ~104 K and is indicated in Fig. 4 as the dashed vertical line (see Methods section for details). This temperature represents the equivalent thermal energy that the ground state of the zero-point vibrational motion of the system contains and for both strain magnitudes, $T_{ZPE}$ is large enough to be in the region of low mechanical reversibility. This implies that zero-point vibration energy is enough to overcome these ripples in the PEL, and the reversible loss will not occur in reality.

## Discussion

The transition we observe in the $\langle f_{lossy}(n, n+1) \rangle$ fraction as a function of temperature shows that only at very low temperatures and in classical simulation, mechanical relaxation occurs within mostly the same groups of atoms and thus is highly reversible. At higher temperatures or if the quantum effect is included, mechanical relaxation occurs through new groups of lossy atoms each cycle, thus relaxation is characterized as irreversible. As temperature increases, so does the accessible relaxation processes in the PEL, leading to transience. This observation is consistent with the view of the PEL describing the 'valleys' of the PEL not as smooth, but rough[7,25]. The complex, fractal-like nature of the PEL basin was shown by Liu et al.[6] where an external shear to an atomistic simulation of a Cu–Zr metallic glass created tortuous pathways for low-barrier activations in the PEL. The possibility of local rearrangements even deep in the glassy state is further supported by the observation that in various metallic glasses the medium range order freezes at the glass transition but the short range order does not[26]. One would imagine that the small local energy barriers may relax out after some time. However, they do not, most likely reflecting the fundamental structural frustration[27,28].

It is interesting to note that the 'roughness' of the PEL allows for dynamics that are of lower energy[29,30] than $\alpha$ and $\beta$ relaxations. This changes our view on the nature of mechanical loss; it is mostly stochastic and irreversible, and not reversible as is often assumed. By using two different strain magnitudes we find two different activation energies of around $E_A \approx 0.6$meV and $E_A \approx 0.1$meV. In metallic glasses, the atomic displacement due to external stress is different for each atom, because all atoms have different local structures. As a result, the PEL is altered when stress is applied. The activation energies that we find at 2.5% and 1% strain reflect these changes. If we assume that the activation energy is equal to the elastic energy due to the applied strain, $E_A \approx (GV/2)\varepsilon_A^2$, where $G$ is shear modulus, $V$ is the local volume of the object and $\varepsilon_A$ is external shear amplitude, we find $V \approx 12^3$, which is comparable to the atomic volume. Thus, this activation is achieved by the motion of a single atom. This is consistent with the observation that about 20% of atoms are liquid-like, and can

easily be activated[31]. In Fig. 1, the population of the lossy atoms with $\delta_{atom} \in [\frac{\pi}{4}, \frac{3\pi}{4}]$ is ~10%. At temperatures above roughly 50 K, these lossy atoms largely revert to the system phase shift distribution in the subsequent cycle as seen in Fig. 2. However, the percentage of these lossy atoms that continue to remain lossy is roughly ~20%, which is a slight increase over the percentage of lossy atoms in the system. This small increase may be a consequence of rejuvenation induced by cyclic deformation. During a deformation event, when the system reaches the saddle point in the PEL the configurational temperature is so high that locally a glass turns into a liquid for a very short time[32]. When the system relaxes from the saddle point to the final state, the relaxation process occurs quickly, in the timescale of 1 ps or less, so that locally liquid is rapidly quenched to a glass. This rapid local cooling results in a state with high fictive temperature, resulting in rejuvenation[33].

Plastic behavior in disordered solids has been extensively studied using athermal quasistatic simulations (AQS)[34,35]. In AQS, small strain steps are applied to glasses at 0 K followed by potential minimization to move the system to a local minimum in the PEL, mimicking the slow timescales of experiments. During these strain steps, there will be occasional plastic atomic rearrangements which have been linked to 'defects' in the structure[36–38]. While the concept of defects has been successful in crystals[39], our result suggests that identifying defects in a glassy structure by AQS simulations may be misleading. Classical AQS simulations do not include any thermal and quantum effects on atomic rearrangements. Therefore, the effectiveness of AQS simulations in glasses is in doubt without assessing these effects. For instance, some of the reversible mechanical relaxation observed in AQS works[40–42] may not be realistic in metallic glasses due to the mechanical and thermal coupling causing stochastic and irreversible behavior shown in this work. Metallic glasses having a transient mechanical relaxation in this timescale is consistent with the observations that periodic loading below the yielding transition can relax or rejuvenate the system[43–45] and it has been used in other works studying internal friction and relaxation[46,47].

Because the energy barriers identified here (<1 meV) are far lower than the zero-point vibrational energy, quantum-mechanically the system does not stay at the energy minimum state, but it is in a quantum resonant state involving multiple local states. Some atoms are rapidly tunneling among two or more sites. Such resonance was observed for superfluid $^4$He by neutron scattering as a peak at 2.3 Å and 0.4 meV[48]. It will be of interest to try to observe such phenomena for metallic glasses, although the signals may be much weaker and more diffuse than in superfluid $^4$He. Quantum resonance is widely known for electrons[49], spins[50], and atoms[51]. The present results suggest that it may be more commonly seen among amorphous materials, particularly in metallic glasses.

In summary, our results demonstrate the complex nature of the underlying PEL in metallic glass which fundamentally determines the relaxation dynamics of the system. The activation energies observed here are much smaller in magnitude in comparison to those for the $\alpha$ and $\beta$ relaxations. They are so small that even quantum mechanical fluctuations could easily overcome these barriers. Thus, the local structure of glass is not static, but is fluctuating, even at room temperature and below. Our results corroborate the experimental studies of low energy mechanical relaxation[14,15] and simulation and theory works showing the fractal nature of the PEL[6,7]. However, our results call into question many of the assumptions made about the nature of metallic glass and challenge the relevance of structural defects and the validity of AQS simulations.

## Methods

### Molecular dynamics simulation
Classical molecular dynamics simulations were performed using the Large-scale Atomic/Molecular Massively Parallel Simulator (LAMMPS)[52]

with a timestep of 1 fs using a $Cu_{64.5}Zr_{35.5}$ metallic glass with 16,000 atoms and a density of 7.84 g/cm³. Each low temperature glass was made by a typical melt-quench method with melting at 3000 K for 1 ns. Four different cooling rates were used: 100K/ps, 1K/ps, 0.1K/ps and 0.01K/ps. The embedded atom method (EAM) potential was used to describe interatomic interactions[53] and periodic boundary conditions were imposed. The relaxed system was then sinusoidally sheared following similar protocols in the literature[46,54] with a sinusoidal strain of $\varepsilon(t) = \varepsilon_A \sin(\omega t)$, where $t$ is time, $\omega = 2\pi/T_\omega$, a period $T_\omega = 100$ ps and two different maximum strains of $\varepsilon_A = 2.5\%$ and $\varepsilon_A = 1.0\%$ to compare impact of strain magnitude. Before collecting data, 30 training/equilibration cycles were done followed by another 30 cycles for data collection.

### Atomic-level loss and lossy fraction
To characterize atomic-level viscoelastic response and identify atoms responsible for mechanical loss, we used atomic-level stresses[55,56] to decompose the system response into atomic-level components[22]. Atomic-level stresses are calculated in LAMMPS as follows:

$$\sigma_i^{ab} = \frac{1}{\Omega_i} \sum_j (f_{ij}^a r_{ij}^b + m_i v_i^a v_i^b) \tag{1}$$

where $a$ and $b$ are Cartesian components, $\Omega_i$ is the atomic volume of atom $i$, $f_{ij}^a$ is the two-body force between atoms $i$ and $j$, $r_{ij}^a$ is the $a$ component of the distance vector, $\mathbf{r}_{ij}$, between atoms $i$ and $j$, and $v_i^a$ is the $a$ component of the velocity of atom $i$. Atomic-level volume is calculated by standard Voronoi tessellation[57]. We identify atoms responsible for mechanical loss by the Fourier transform[58] of the time dependent atomic-level shear stress response seen in Fig. 1a, b. Each atom has a corresponding phase shift ranging from $[-\pi, +\pi]$ and a representative distribution of the system is seen in Fig. 1c. In a previous work[22] we showed atoms with $\delta_{atom} \approx \frac{\pi}{2}$ as responsible for mechanical loss and thus we define 'lossy' atoms in this work as $\delta_{atom} \in [\frac{\pi}{4}, \frac{3\pi}{4}]$ and are represented by the hatched region in Fig. 1c. The temperature scaling of our results is not sensitive to the choice of the boundaries as shown in Supplementary Fig. 5.

The lossy fraction represents the fraction of atoms identified as lossy in some cycle $n$ which remain lossy some cycle $n + m$ later. This is calculated as follows:

$$\langle f_{lossy}(n, n+m) \rangle = \left\langle \frac{N_{lossy,remain}(n, n+m)}{N_{lossy}(n)} \right\rangle \tag{2}$$

This lossy fraction represents number of lossy atoms from cycle ($n$) which remain lossy at a later cycle ($n + m$) with $\langle \cdots \rangle$ representing the average over 20 cycles. This lossy fraction was computed for all four cooling rates and temperatures for a strain magnitude of $\varepsilon_A = 2.5\%$ and is shown in Fig. 3.

### Zero-point energy calculation
We estimate the effective zero-point motion temperature from the vibrational density of states (vDOS) given by the Fourier transform of the velocity autocorrelation function by the following equation:

$$g(\omega) = \sum_{m=1}^{3N_{atom}} \delta(\omega - \omega_m) = \frac{1}{k_B T} \int_0^\infty \sum_{n=1}^{N_{atom}} m_n \langle \mathbf{v}_n(t) \cdot \mathbf{v}_n(0) \rangle e^{i\omega t} dt \tag{3}$$

where $N_{atom}$ is the number of atoms, $m_n$ is the mass of atom $n$, $\mathbf{v}_n(t)$ is the velocity of atom $n$ at some time $t$, thus the quantity $\langle \mathbf{v}_n(t) \cdot \mathbf{v}_n(0) \rangle$ represents the autocorrelation function for atom $n$[59–61]. For better statistics, vDOS calculations were done using five different initial velocity trajectories at 0.1 K. A plot of the vDOS can be found in Supplementary Fig. 6. The effective temperature of the zero-point

vibration energy, $T_{ZPE}$, can then be approximated from the vibrational density of states by:

$$E_0 = \int \frac{1}{2}\hbar\omega g(\omega)d\omega = 3N_{atom}k_B T_{ZPE} \qquad (4)$$

## Data availability
All data are available from the corresponding authors on request.

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

## Acknowledgements

This research was supported by the U.S. Department of Energy, Office of Science, Basic Energy Sciences, Materials Science and Engineering Division. This work used the Extreme Science and Engineering Discovery Environment Expanse under Allocation No. TG-MAT200012. This research used resources of the National Energy Research Scientific Computing Center (NERSC), a U.S. Department of Energy Office of Science User Facility located at Lawrence Berkeley National Laboratory, operated under Contract No. DE-AC02-05CH11231 using NERSC award BES-ERCAP0017758. This manuscript has been authored by UT-Battelle, LLC under Contract No. DE-AC05-00OR22725 with the U.S. Department of Energy. The United States Government retains and the publisher, by accepting the article for publication, acknowledges that the United States Government retains a non-exclusive, paid-up, irrevocable, world-wide license to publish or reproduce the published form of this manuscript, or allow others to do so, for United States Government purposes. The Department of Energy will provide public access to these results of federally sponsored research in accordance with the DOE Public Access Plan (http://energy.gov/downloads/doe-public-access-plan).

## Author contributions

T.E. conceived the research. L.Z. performed simulations and data analysis with contributions from J.M. L.Z, J.M. and T.E. interpreted the results. L.Z. wrote the paper with contributions from J.M and T.E.

## Competing interests

The authors declare no competing interests.
