## [Peer Review File · Nature Communications]

Ripples in the Bottom of the Potential Energy Landscape of Metallic GlassREVIEWER COMMENTS

Reviewer #1 (Remarks to the Author):

This paper looks at simulations of periodical sheared metallic glass where they specifically consider an experimental realizable system of CuZr.

The authors then look at the energy barriers for atoms to jump, which can be very small, down to 1 meV, so even system quantum effects could play a role. The authors then argue that this implies that metallic glass is never totally frozen, at least down at the local atomic level.

In principle, this is an interesting result and has implications for metallic glasses and quantum effects in glasses in general. Still, I have some questions about the systems and what could be seen in the experiment. They do an excellent job looking at different temperatures, but I did not see much on changing AC cycle amplitude or frequency, which will be very important. Below, I highlight the issues.

(1) For the first step, they should have actual images of the system and say the energy per cycle to see if the system settles into a reversible state or if the energy drops over time. Also, they should look more at how this changes with the cycle amplitude and frequency. It could anneal out some low-energy barriers.

(2) Does it matter how the system is prepared?

It seems like one could prepare the system way where the barriers are more profound. It is known that glassy materials, including metallic glasses can have very different shear responses or yield points depending on how glasses are prepared, such as fragile versus strong glass. I would not be surprised if it is possible to prepare glass in a high-energy metastable state that would have small barriers for atoms to jump to many different states. Below, I highlight the issues. Also, are they only doing up to eight cycles?

(3) Related to the first one, if the authors wait enough time at even low temperatures before cycling, the local energy barriers change, and the small barriers vanish. On the other hand, one meV barrier between states suggests something like spin ice, where the system is frustrated and has multiple states that it could readily jump to. In many ways, frustrated systems are liquid even at $T = 0.0$ with excess entropy. The authors could calculate the excess entropy.

Zero-point entropy in 'spin ice'

A. P. Ramirez, A. Hayashi, R. J. Cava, R. Siddharthan & B. S. Shastry
Nature volume 399, pages333–335 (1999)Cite this article

(4) Could the authors say something more about the experimental implications of this. This could even be speculative For example, if the quantum effects are reverent, one could imagine the "quantum creep" of soft spots in this system. I think one could exert a strain on the system and look at creep, As the temperatures are lowered at some point, the creep will

saturate once the thermal creep level is below the quantum creep level.
In principle, quantum creep should be possible in these since it is the scale of atoms and the barriers are small. Quantum creep of larger objects like vortices in type-II superconductors, CDWs, and solid Helium has been discussed before. This could help with the higher impact needed for NatureCommunications.

Quantum collective creep
Gianni Blatter, Vadim B. Geshkenbein, and Valeri M. Vinokur
Phys. Rev. Lett. 66, 3297 – Published 24 June 1991

Tunneling of a quantized vortex: Roles of pinning and dissipation
Ping Ao and David J. Thouless
Phys. Rev. Lett. 72 132 (1994)

Macroscopic Quantum Tunneling in Quasi One-Dimensional Metals. II. Theory
John Bardeen
Phys. Rev. Lett. 55 1010 (1985)

General Features of Quantum Creep in High-TcSuperconductors
A. F. Th. Hoekstra, R. Griessen, A. M. Testa, J. el Fattahi, M. Brinkmann, K. Westerholt, W. K. Kwok, and G. W. Crabtree
Phys. Rev. Lett. 80 4293 (1998)

Giant Plasticity of a Quantum Crystal
Ariel Haziot, Xavier Rojas, Andrew D. Fefferman, John R. Beamish, and Sébastien Balibar
Phys. Rev. Lett. 110, 035301 2013

Quantum fluctuations can promote or in
Thomas E. Markland, Joseph A. Morrone, Bruce J. Berne, Kunimasa Miyazaki, Eran Rabani & David R. Reichman
Nature Physics volume 7, pages134–137 (20

Reviewer #2 (Remarks to the Author):

Authors reported a simulation calculation and analysis study on lossy atom distribution, temperature dependence of lossy fraction and effective temperature of the zero-point motion in metallic glass below the temperature 300 K during mechanical relaxation by utilizing molecular dynamics dynamic mechanical spectroscopy (MD-DMS) on a prototypical CuZr alloy. The main intrigue results in the work are such as: they find that above 100 K, most lossy atoms identified in one shear deformation cycle convert to non-lossy behavior in the next cycle, demonstrating high transience of local atomic structure participating in mechanical relaxation. In contrast, at cryogenic temperatures lossy atoms are largely reversible. They also give an activation energy of ~ 0.6 meV at a strain of 2.5 % and ~ 0.1 meV at a strain of 1 %. The activation energies observed are much smaller in magnitude in

comparison to those for the α and β relaxations, which is ascribed to the small ripples in the bottom of the basin in the PEL. They estimate effective temperature of the zero-point motion, TZPE to be ~ 104 K, which is large enough to be in the region of low mechanical reversibility. These results suggest that the atoms in system moves easily within the PEL basin through quantum fluctuations and thermal excitations and this seems to challenge many assumptions regarding the thermodynamic states of metallic glasses. This work can be considered to publish in the NCOMMS. But before accepted the issues need to be clarified or explained more clearly.

1. The activation energy is the minimum amount of extra energy that is required to activate atoms or molecules to a condition in which they can undergo chemical transformation or physical transport. Therefore, activation energy is an intrinsic property of matter. However, authors give an activation energy of ~ 0.6 meV at a strain of 2.5 % and ~ 0.1 meV at a strain of 1 % for the studied glass system. This means that the activation energy changes with the applied magnitude of the strain, or there is an activation energy spectrum corresponding this strain. How can this change be explained? If so, whether this means that the applicability of the exponential fitting function used by the authors in this work is debatable.

2. In the last paragraph of introduction section, authors said that "we estimate the magnitude of quantum fluctuations using ...", implying that quantum fluctuations easily connect multiple subbasins", but authors didn't directly show the magnitude of quantum fluctuations at 0.1 K or at lower temperature in the text or supplementary information, and only an effective temperature of the zero-point motion, TZPE (to be ~ 104 K) is given according to equation (4) ($E_0 = 3N_{atom}k_B T_{ZPE}$). However, equation (4) generally is used for describe the relation of the average kinetic energy per particle (such as atom or molecular) with temperature. The zero-point energy is the energy that atoms or molecules retain even at the absolute zero of temperature. In the case, the best way is to directly give the estimated zero-point energy of the system at or near 0 K temperature in the text in order to intuitively see that the magnitude of this quantum fluctuations is enough to overcome these potential barriers.

3. In the picture of quantum mechanical zero-point energy fluctuation, zero-point energy is the energy that atoms or molecules retain even at the absolute zero of temperature, but authors mentioned the concept of the effective temperature of the zero-point motion energy and didn't give a further explanation to this. By the way, the movement involved in the text is vibration, so it is suggested that the word "zero-point vibration energy" should be used uniformly through the paper.

4. There is a writing error in the unit dimension of the local volume, V in the last line of page7.

Response to the reviewers for the paper entitled

“Ripples in the Bottom of the Potential Energy Landscape of Metallic Glass”

Ms. No. NCOMMS-23-34243

The authors thank the reviewers for their thoughtful comments and suggestions. The paper has been revised to reflect these comments. A detailed revision/rebuttal is documented below. Before responding to each individual’s comments, we would like to first clarify our new findings of this work.

This work provides crucial new insights into the potential energy landscape of metallic glass which is of great importance for understanding the dynamic phenomena that exist in metallic glass. We, therefore, believe that this work meets the criteria of broad interest and impact for publication at *Nature Communications*.

Detailed response to Reviewer #1:

1. *For the first step, they should have actual images of the system and say the energy per cycle to see if the system settles into a reversible state or if the energy drops over time. Also, they should look more at how this changes with the cycle amplitude and frequency. It could anneal out some low-energy barriers.*

Response: Due to the low energy scale of the associated rearrangements and the small activation volume, the atomic rearrangements are not immediately obvious through visualization of the system. We find that through the shear strain cycling, the energy of the system slightly goes down in the early cycles then settles into a mechanical equilibrium where the system potential energy remains roughly the same, as is seen in Fig. S4. We also found there was essentially no difference in change of potential energy between $\varepsilon_A = 2.5\%$ versus $\varepsilon_A = 1.0\%$. Regarding the importance of cycle frequency, other literature has shown that this regime of fast relaxation has a relatively weak dependence on frequency and temperature, being sandwiched between the beta-relaxation and the boson-peak¹. We are only likely to see large changes in mechanical response with frequencies that are many orders of magnitude in difference, which is not accessible with simulation due to high computational costs for long simulation.

2. *Does it matter how the system is prepared? It seems like one could prepare the system way where the barriers are more profound. It is known that glassy materials, including metallic glasses can have very different shear responses or yield points depending on how glasses are prepared, such as fragile versus strong glass. I would not be surprised if it is possible to prepare glass in a high-energy metastable state that would have small barriers for atoms to jump to many different states. Below, I highlight the issues. Also, are they only doing up to eight cycles?*

Response: Indeed, the glass structure depends on how it was prepared, such as the rate of cooling from the liquid. We used 4 systems prepared with different cooling rates. We have found that cooling rate had minimal impact on the characteristic response of the lossy fraction as seen in Fig. 4. The fastest cooling rate of 100 K/ps is known to result in strongly modified distribution of energy barriers². The lack of difference in the behavior of the lossy fraction of atoms versus cooling rate may be in part due to the mechanical conditioning that is performed through the 30 cycles of shear straining before analyzing mechanical loss. The effect of the mechanical conditioning can be seen in the relatively rapid reduction of potential energy per atom as shown in Fig. S4. Some clarification on the number of cycles was added to the manuscript.

New (page 5, line 8): Over the 30 cycles in which lossy atoms were identified,

3. *Related to the first one, if the authors wait enough time at even low temperatures before cycling, the local energy barriers change, and the small barriers vanish. On the other hand, one meV barrier between states suggests something like spin ice, where the system is frustrated and has multiple states that it could readily jump to. In many ways, frustrated systems are liquid even at $T = 0.0$ with excess entropy. The authors could calculate the excess entropy.*

Response: One would imagine that the small local energy barriers may relax out after some time. However, they do not, most likely reflecting the fundamental structural frustration³⁻⁵ as the reviewer correctly pointed out. We still see these small barriers even at the slowest cooling rate; they are likely to be a persistent feature at the bottom of the sub-basins in the PEL. We added a paragraph to discuss this point, and we thank the reviewer for suggesting this subject. Now, excess entropy is quite difficult to calculate, particularly for metallic systems. In ionic or covalent glasses, the configurational entropy may be specified via bond counting. However, in metallic glasses atomic bonds are not clearly defined due to the nature of metallic bonding. There are some measures such as the two-body excess entropy⁶, but neglecting higher order terms could be problematic⁷. In addition, it is not clear what the physical meaning of this two-body excess entropy is.

New (page 7, line 19): One would imagine that the small local energy barriers may relax out after some time. However, they do not, most likely reflecting the fundamental structural frustration.

4. *Could the authors say something more about the experimental implications of this. This could even be speculative. For example, if the quantum effects are relevant, one could imagine the "quantum creep" of soft spots in this system. I*

think one could exert a strain on the system and look at creep, As the temperatures are lowered at some point, the creep will saturate once the thermal creep level is below the quantum creep level. In principle, quantum creep should be possible in these since it is the scale of atoms and the barriers are small. Quantum creep of larger objects like vortices in type-II superconductors, CDWs, and solid Helium has been discussed before. This could help with the higher impact needed for NatureCommunications.

Response: When the magnitude of the zero-point energy is less than the barrier height cooperative quantum creep (tunneling) can occur. This has been discussed by many, including one of the authors 50 years ago⁸. However, here the zero-point energy (27 meV) is larger than the barrier height. Thus, the system is in the quantum resonant state involving multiple atomic sites. We included a sentence to discuss this point.

New (page 9, line 16): Because the energy barriers identified here (< 1 meV) are far lower than the zero-point vibrational energy, quantum-mechanically the system does not stay at the energy minimum state, but it is in a quantum resonant state involving multiple local states. Some atoms are rapidly tunneling among two or more sites. Such resonance was observed for superfluid ⁴He by neutron scattering as a peak at 2.3 Å and 0.4 meV⁴⁸. It will be of interest to try to observe such phenomena for metallic glasses, although the signals may be much weaker and more diffuse than in superfluid ⁴He. Quantum resonance is widely known for electrons⁴⁹, spins⁵⁰, and atoms⁵¹. The present results suggest that it may be more commonly seen among amorphous materials, particularly in metallic glasses.

Detailed response to Reviewer #2:

1. *The activation energy is the minimum amount of extra energy that is required to activate atoms or molecules to a condition in which they can undergo chemical transformation or physical transport. Therefore, activation energy is an intrinsic property of matter. However, authors give an activation energy of ~0.6 meV at a strain of 2.5 % and ~0.1 meV at a strain of 1 % for the studied glass system. This means that the activation energy changes with the applied magnitude of the strain, or there is an activation energy spectrum corresponding this strain. How can this change be explained? If so, whether this means that the applicability of the exponential fitting function used by the authors in this work is debatable.*

Response: Conventionally, glasses are described as sitting in a meta-stable potential. Therefore, depending on how glasses are prepared (e.g., different cooling rates), activation energies of structural relaxation processes will vary. Further, in metallic glass the atomic displacement due to external stress is different

for each atom, because all atoms have different local structures. As a result, the PEL is altered when stress is applied. The activation energies that we find at 2.5 % and 1% strain reflect these changes. In crystalline solids the structure is merely affinely distorted, thus the concept of the activation volume applies, but that is not the case for metallic glasses.

New (page 8, line 5): In metallic glass the atomic displacement due to external stress is different for each atom, because all atoms have different local structures. As a result, the PEL is altered when stress is applied. The activation energies that we find at 2.5 % and 1% strain reflect these changes.

- In the last paragraph of introduction section, authors said that “we estimate the magnitude of quantum fluctuations using ..., implying that quantum fluctuations easily connect multiple subbasins”, but authors didn’t directly show the magnitude of quantum fluctuations at 0.1 K or at lower temperature in the text or supplementary information, and only an effective temperature of the zero-point motion, T_{ZPE} (to be ~ 104 K) is given according to equation (4) ($E_0 = 3N_{atom}k_B T_{ZPE}$). However, equation (4) generally is used for describing the relation of the average kinetic energy per particle (such as atom or molecular) with temperature. The zero-point energy is the energy that atoms or molecules retain even at the absolute zero of temperature. In this case, the best way is to directly give the estimated zero-point energy of the system at or near 0 K temperature in the text in order to intuitively see that the magnitude of this quantum fluctuations is enough to overcome these potential barriers.*

Response: We thank the reviewer for bringing up the point on the magnitude of quantum fluctuations. We have added in the manuscript the calculated vibrational ground state energy per atom. Regarding the estimated zero-point vibrational energy, we calculate essentially what is the equivalent thermal energy to the calculated zero-point vibrational energy. This was done so that we could easily compare the energy scale of the zero-point vibrational energy to the temperature dependence of the lossy fraction calculation.

New (page 6, line 21): we calculate the total ground state energy to be 26.8 meV/atom

- In the picture of quantum mechanical zero-point energy fluctuation, zero-point energy is the energy that atoms or molecules retain even at the absolute zero of temperature, but authors mentioned the concept of the effective temperature of the zero-point motion energy and didn’t give a further explanation to this. By the way, the movement involved in the text is vibration, so it is suggested that the word “zero-point vibration energy” should be used uniformly through the paper.*

Response: We have added further detail in the manuscript on the meaning of effective temperature in this context. Additionally, we have taken the reviewers suggestion and used “zero-point vibration energy” for clarity throughout the manuscript.

New (page 7, line 2): This temperature represents the equivalent thermal energy that the ground state of the zero-point vibrational motion of the system contains...

New (page 6, line 21): By equating this ground state energy to the total thermal energy of the system we calculate an

4. *There is a writing error in the unit dimension of the local volume, V in the last line of page7.*

Response: We have corrected the mistake and modified the manuscript.

References

1. Yu, H.-B. & Samwer, K. Atomic mechanism of internal friction in a model metallic glass. *Phys. Rev. B* **90**, 144201 (2014).
2. Fan, Y., Iwashita, T. & Egami, T. Energy landscape-driven non-equilibrium evolution of inherent structure in disordered material. *Nature Communications* **8**, 1–7 (2017).
3. Nelson, D. R. Order, frustration, and defects in liquids and glasses. *Phys. Rev. B* **28**, 5515–5535 (1983).
4. Sethna, J. P. Frustration and Curvature: Glasses and the Cholesteric Blue Phase. *Phys. Rev. Lett.* **51**, 2198–2201 (1983).
5. Egami, T., Levashov, V., Aga, R. & Morris, J. R. Geometrical Frustration and Glass Formation. *Metall Mater Trans A* **39**, 1786–1790 (2008).
6. Rosenfeld, Y. Relation between the transport coefficients and the internal entropy of simple systems. *Phys. Rev. A* **15**, 2545–2549 (1977).
7. Wallace, D. C. On the role of density fluctuations in the entropy of a fluid. *The Journal of Chemical Physics* **87**, 2282–2284 (1987).
8. Egami, T. Theory of Bloch wall tunnelling. *physica status solidi (b)* **57**, 211–224 (1973).
48. Dmowski, W. *et al.* Observation of dynamic atom-atom correlation in liquid helium in real space. *Nat Commun* **8**, 15294 (2017).
49. Pauling, L. *The nature of the chemical bond and the structure of molecules and crystals: an introduction to modern structural chemistry.* (Cornell Univ. Press, 2010).
50. Anderson, P. W. *The theory of superconductivity in the high-T_c cuprates.* (Princeton University Press, 1997).

51. Müller, K. A. & Burkard, H. SrTiO₃: An intrinsic quantum paraelectric below 4 K.

Phys. Rev. B **19**, 3593–3602 (1979).

REVIEWERS' COMMENTS

Reviewer #1 (Remarks to the Author):

The authors have replied to my comments and made several changes which I am now satisfied with. The authors have also made some changes to address the point of the other referee and overall the paper is much improved. I now recommend for publication.

Reviewer #2 (Remarks to the Author):

Authors answered those questions from reviewers and make an appropriate revisions in text. So I think the manuscript can be accepted for publication in NComms.